# TAN: A Transferable Adversarial Network for DNN-Based UAV SAR Automatic Target Recognition Models

**Meng Du** [1], **Yuxin Sun** [2], **Bing Sun** [3], **Zilong Wu** [1], **Lan Luo** [4], **Daping Bi** [1] **and Mingyang Du** [1,*]

1   College of Electronic Engineering, National University of Defense Technology, Hefei 230037, China
2   Science and Technology on Electro-Optical Information Security Control Laboratory, Tianjin 300308, China
3   China Satellite Maritime Tracking and Control Department, Jiangyin 214430, China
4   College of Communication Engineering, Lanzhou University, Lanzhou 730030, China
*   Correspondence: dumingyang17@nudt.edu.cn

**Abstract:** Recently, the unmanned aerial vehicle (UAV) synthetic aperture radar (SAR) has become a highly sought-after topic for its wide applications in target recognition, detection, and tracking. However, SAR automatic target recognition (ATR) models based on deep neural networks (DNN) are suffering from adversarial examples. Generally, non-cooperators rarely disclose any SAR-ATR model information, making adversarial attacks challenging. To tackle this issue, we propose a novel attack method called Transferable Adversarial Network (TAN). It can craft highly transferable adversarial examples in real time and attack SAR-ATR models without any prior knowledge, which is of great significance for real-world black-box attacks. The proposed method improves the transferability via a two-player game, in which we simultaneously train two encoder–decoder models: a generator that crafts malicious samples through a one-step forward mapping from original data, and an attenuator that weakens the effectiveness of malicious samples by capturing the most harmful deformations. Particularly, compared to traditional iterative methods, the encoder–decoder model can one-step map original samples to adversarial examples, thus enabling real-time attacks. Experimental results indicate that our approach achieves state-of-the-art transferability with acceptable adversarial perturbations and minimum time costs compared to existing attack methods, making real-time black-box attacks without any prior knowledge a reality.

**Keywords:** unmanned aerial vehicle (UAV); synthetic aperture radar (SAR); automatic target recognition (ATR); deep neural network (DNN); adversarial example; transferability; encoder–decoder; real-time attack

## 1. Introduction

The ongoing advances in unmanned aerial vehicle (UAV) and synthetic aperture radar (SAR) technologies have enabled the acquisition of high-resolution SAR images through UAVs. However, unlike visible light imaging, SAR images reflect the reflection intensity of imaging targets to radar signals, making it difficult for humans to extract effective semantic information from SAR images without the aid of interpretation tools. Currently, deep learning has achieved excellent performance in various scenarios [1–3], and SAR automatic target recognition (SAR-ATR) models based on deep neural networks (DNN) [4–8] have become one of the most popular interpretation methods. With their powerful representation capabilities, DNNs outperform traditional approaches in image classification tasks. However, recent studies have shown that DNN-based SAR-ATR models are susceptible to adversarial examples [9].

The concept of adversarial examples was first proposed by Szegedy et al. [10], which suggests that a carefully designed tiny perturbation can cause a well-trained DNN model to misclassify. This finding has made adversarial attacks one of the most serious threats to artificial intelligence (AI) security. To date, researchers have proposed a variety of adversarial

attack methods, which can be mainly divided into two categories from the perspective of prior knowledge: the white-box and black-box attacks. In the first case, attackers can utilize a large amount of prior knowledge, such as the model structure and gradient information, etc., to craft adversarial examples for victim models. Examples of white-box methods include gradient-based attacks [11,12], boundary-based attacks [13], and saliency map-based attacks [14], etc. In the second case, attackers can only access the output information or even less, making adversarial attacks much more difficult. Examples of black-box methods include probability label-based attacks [15,16] and decision-based attacks [17], etc. We now consider an extreme situation, where attackers have no access to any feedback from victim models, such that existing attack methods are unable to craft adversarial examples until researchers discover that adversarial examples can transfer among DNN models performing the same task [18]. Recent relevant studies focused on improving the basic FGSM [11] method to enhance the transferability of adversarial examples, such as gradient-based methods [19,20], transformation-based methods [20,21], and variance-based methods [22], etc. However, the transferability and real-time performance of the above approaches are still insufficient to meet realistic attack requirements. Consequently, adversarial attacks are pending further improvements.

With the wide application of DNNs in the field of remote sensing, researchers have embarked on investigating the adversarial examples of remote sensing images. Xu et al. [23] first investigated the adversarial attack and defense in safety-critical remote sensing tasks, and proposed the mixup attack [24] to generate universal adversarial examples for remote sensing images. However, the research on the adversarial example of SAR images is still in its infancy. Li et al. [25] generated abundant adversarial examples for CNN-based SAR image classifiers through the basic FGSM method and systematically evaluated critical factors affecting the attack performance. Du et al. [26] designed a Fast C&W algorithm to improve the efficiency of generating adversarial examples by introducing an encoder–decoder model. To enhance the universality and feasibility of adversarial perturbations, the work in [27] presented a universal local adversarial network to generate universal adversarial perturbations for the target region of SAR images. Furthermore, the latest research [28] has broken through the limitations of the digital domain and implemented the adversarial example of SAR images in the signal domain by transmitting a two-dimensional jamming signal. Despite the high attack success rates achieved by the above methods, the problem of transferable adversarial examples in the field of SAR-ATR has yet to be addressed.

In this paper, a transferable adversarial network (TAN) is proposed to improve the transferability and real-time performance of adversarial examples in SAR images. Specifically, during the training phase of TAN, we simultaneously trained two encoder–decoder models: a generator that crafts malicious samples through a one-step forward mapping from original data, and an attenuator that weakens the effectiveness of malicious samples by capturing the most harmful deformations. We argue that if the adversarial examples crafted by the generator are robust to the deformations produced by the attenuator, i.e., the attenuated adversarial examples remain effective to DNN models, then they are capable of transferring to other victim models. Moreover, unlike traditional iterative methods, our approach can one-step map original samples to adversarial examples, thus enabling real-time attacks. In other words, we realize real-time transferable adversarial attacks through a two-player game between the generator and attenuator.

The main contributions of this paper are summarized as follows.

(1) For the first time, this paper systematically evaluates the transferability of adversarial examples among DNN-based SAR-ATR models. Meanwhile, our research reveals that there may be potential common vulnerabilities among DNN models performing the same task.

(2) We propose a novel network to enable real-time transferable adversarial attacks. Once the proposed network is well-trained, it can craft adversarial examples with high transferability in real time, thus attacking black-box victim models without resorting to any prior knowledge. As such, our approach possesses promising applications in AI security.

(3) The proposed method is evaluated on the most authoritative SAR-ATR dataset. Experimental results indicate that our approach achieves state-of-the-art transferability with acceptable adversarial perturbations and minimum time costs compared to existing attack methods, making real-time black-box attacks without any prior knowledge a reality.

The rest parts of this paper are arranged as follows. Section 2 introduces the relevant preparation knowledge, and Section 3 describes the proposed method in detail. The experimental results and conclusions are given in Sections 4 and 5, respectively.

## 2. Preliminaries

### 2.1. Adversarial Attacks for DNN-Based SAR-ATR Models

Suppose $x_n \in [0, 255]^{W \times H}$ is a single channel SAR image from the dataset $\mathcal{X}$ and $f(\cdot)$ is a DNN-based $K$-class SAR-ATR model. Given a sample $x_n$ as input to $f(\cdot)$, the output is a $K$-dimensional vector $f(x_n) = [f(x_n)_1, f(x_n)_2, \cdots, f(x_n)_K]$, where $f(x_n)_i \in \mathbb{R}$ denotes the score of $x_n$ belonging to class $i$. Let $C_p = \arg\max_i(f(x_n)_i)$ represent the predicted class of $f(\cdot)$ for $x_n$. The adversarial attack is to fool $f(\cdot)$ with an adversarial example $\tilde{x}_n$ that only has a minor perturbation on $x_n$. The detail process can be expressed as follows:

$$\arg\max_i f(\tilde{x}_n)_i \neq C_p, \quad \text{s.t.} \|\tilde{x}_n - x_n\|_p \leq \xi \tag{1}$$

where the $L_p$-norm is defined as $\|v\|_p = (\sum_i |v_i|^p)^{\frac{1}{p}}$, and $\xi$ controls the magnitude of adversarial perturbations. The common $L_p$-norm includes the $L_0$-norm, $L_2$-norm, and $L_\infty$-norm. Attackers can select different norm types according to practical requirements. For example, the $L_0$-norm represents the number of modified pixels in $\tilde{x}_n$, the $L_2$-norm measures the mean square error (MSE) between $\tilde{x}_n$ and $x_n$ and the $L_\infty$-norm denotes the maximum variation for individual pixels in $\tilde{x}_n$.

Meanwhile, adversarial attacks can be mainly divided into two modes. The first basic mode is called the non-targeted attack, making DNN models misclassify. The second one is more stringent, called the targeted attack, which induces models to output specified results. There is no doubt that the latter poses a higher level of threat to AI security. In other words, the non-targeted attack is to minimize the probability of models correctly recognizing samples; conversely, the targeted attack maximizes the probability of models identifying samples as target classes. Thus, (1) can be transformed into the following optimization problems:

- For the non-targeted attack:

$$minimize(\frac{1}{N}\sum_{n=1}^{N} D(\arg\max_i f(\tilde{x}_n)_i == C_{tr})), \quad \text{s.t.} \|\tilde{x}_n - x_n\|_p \leq \xi \tag{2}$$

- For the targeted attack:

$$maximize(\frac{1}{N}\sum_{n=1}^{N} D(\arg\max_i f(\tilde{x}_n)_i == C_{ta})), \quad \text{s.t.} \|\tilde{x}_n - x_n\|_p \leq \xi \tag{3}$$

where the discriminant function $D(\cdot)$ equals one if the equation holds; otherwise, it equals zero. $C_{tr}$ and $C_{ta}$ represent the true and target classes of the input. $N$ is the number of samples in the dataset. Obviously, the above optimization problems are exactly the opposite of a DNN's training process, and the corresponding loss functions will be given in the next chapter.

### 2.2. Transferability of Adversarial Examples

We consider an extreme situation where attackers have no access to any feedback from victim models, in which existing white-box and black-box attacks are unable to craft adversarial examples. In this case, attackers can utilize the transferability of adversarial examples to attack models. Specifically, the extensive experiments in [18] have demonstrated that adversarial examples can transfer among models, even if they have different architectures or are trained on different training sets, so long as they are trained to perform the same task. Details about the transferability are shown in Figure 1.

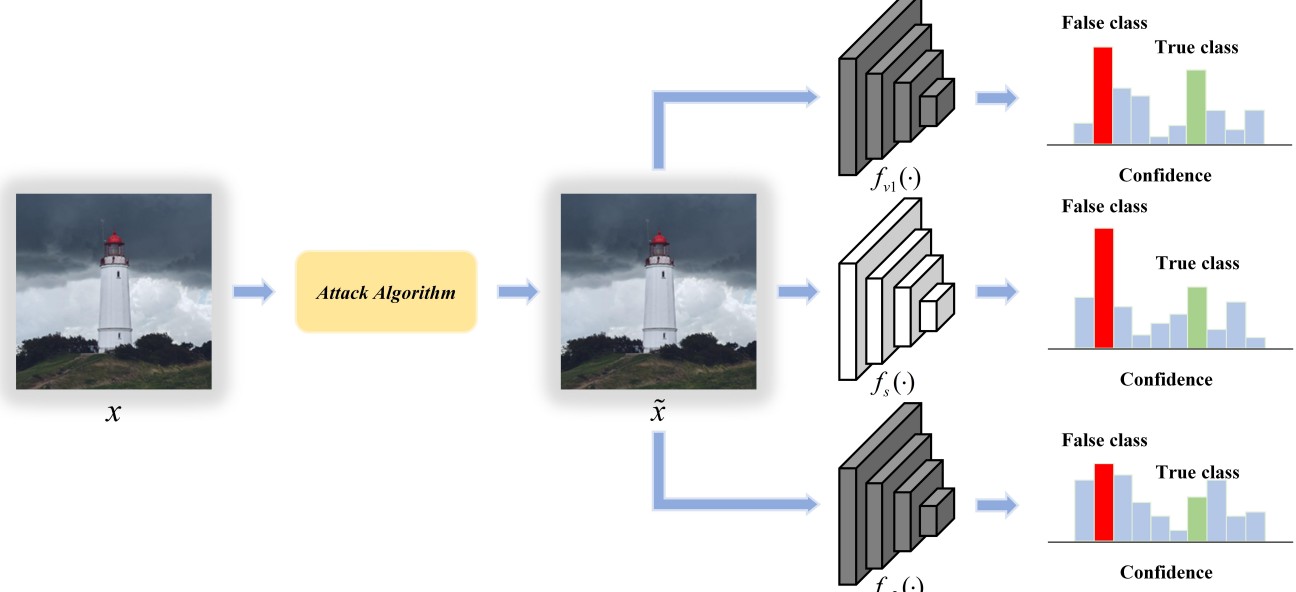

**Figure 1.** Transferability of adversarial examples.

As shown in Figure 1, for an image classification task, we have trained three recognition models. Suppose that only the surrogate model $f_s(\cdot)$ is a white-box model, and victim models $f_{v1}(\cdot)$, $f_{v2}(\cdot)$ are black-box models. Undoubtedly, given an sample $x$, attackers can craft an adversarial example $\tilde{x}$ to fool $f_s(\cdot)$ through attack algorithms. Meanwhile, given the transferability of adversarial examples, $\tilde{x}$ can also fool $f_{v1}(\cdot)$ and $f_{v2}(\cdot)$ successfully. However, the transferability generated by existing algorithms is very weak, so this paper is dedicated to crafting highly transferable adversarial examples.

### 3. The Proposed Transferable Adversarial Network (TAN)

In this paper, the proposed Transferable Adversarial Network (TAN) utilizes the encoder–decoder model and data augmentation technology to improve the transferability and real-time performance of adversarial examples. The framework of our network is shown in Figure 2. As we can see, compared to traditional iterative methods, TAN introduces a generator $G(\cdot)$ to learn the one-step forward mapping from the clean sample $x$ to the adversarial example $\tilde{x}$, thus enabling real-time attacks. Meanwhile, to improve the transferability of $\tilde{x}$, we simultaneously trained an attenuator $A(\cdot)$ to capture the most harmful deformations, which are supposed to weaken the effectiveness of $\tilde{x}$ while still preserving the semantic meaning of $x$. We argue that if $\tilde{x}$ is robust to the deformations produced by $A(\cdot)$, i.e., $\tilde{x}^*$ remains effective to DNN models, then $\tilde{x}$ is capable of transferring to the black-box victim model $f_v(\cdot)$. In other words, we achieve real-time transferable adversarial attacks through a two-player game between $G(\cdot)$ and $A(\cdot)$. This chapter will introduce our method in detail.

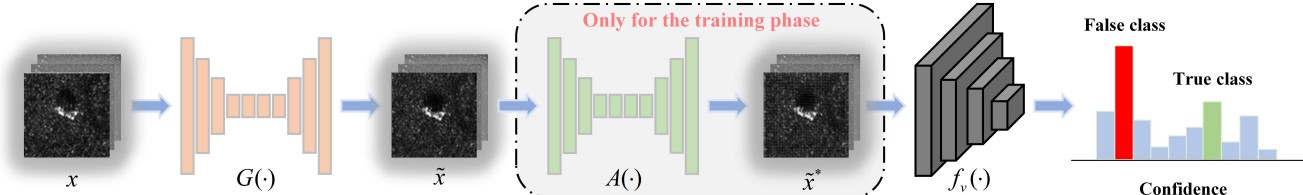

**Figure 2.** Framework of TAN.

### 3.1. Training Process of the Generator

For easy understanding, Figure 3 shows the detailed training process of the generator. Note that a white-box model is selected as the surrogate model $f_s(\cdot)$ during the training phase.

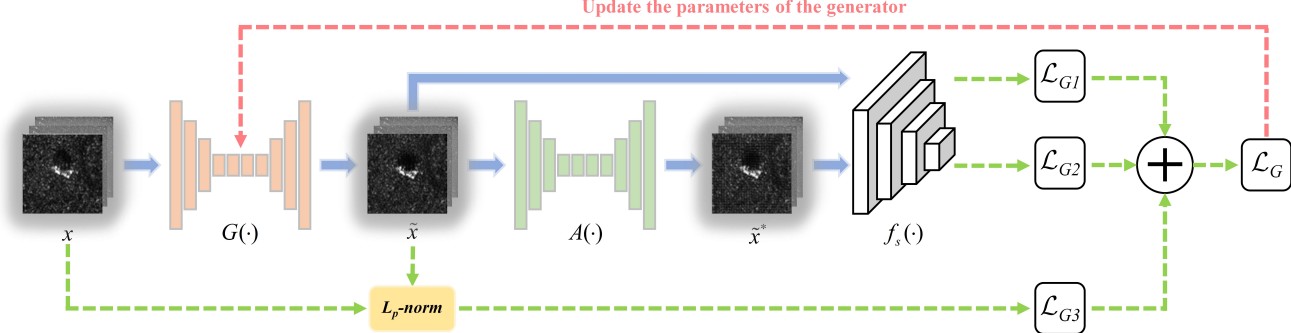

**Figure 3.** Training process of the generator.

As we can see, given a clean sample $x$, the generator $G(\cdot)$ crafts the adversarial example $\tilde{x}$ through a one-step forward mapping, as follows:

$$\tilde{x} = G(x) \tag{4}$$

Meanwhile, the attenuator $A(\cdot)$ takes $\tilde{x}$ as input and outputs the attenuated adversarial example $\tilde{x}^*$:

$$\tilde{x}^* = A(\tilde{x}) \tag{5}$$

Since $\tilde{x}$ has to fool $f_s(\cdot)$ with a minor perturbation, and $\tilde{x}^*$ needs to remain effective against $f_s(\cdot)$, the loss function of $G(\cdot)$ consists of three parts. Next, we will give the generator loss $\mathcal{L}_G$ of non-targeted and targeted attacks, respectively.

For the non-targeted attack: First, according to (2), $\tilde{x}$ is to minimize the classification accuracy of $f_s(\cdot)$, which means that it has to decrease the confidence of being recognized as the true class $C_{tr}$, i.e., to increase the confidence of being identified as others. Thus, the first part of $\mathcal{L}_G$ can be expressed as:

$$
\begin{aligned}
\mathcal{L}_{G1}(f_s(\tilde{x}), C_{tr}) &= -\log\left(\frac{\sum_{i \neq C_{tr}} \exp(f_s(\tilde{x})_i)}{\sum_i \exp(f_s(\tilde{x})_i)}\right) \\
&= -\log\left(1 - \frac{\exp(f_s(\tilde{x})_{C_{tr}})}{\sum_i \exp(f_s(\tilde{x})_i)}\right)
\end{aligned}
\tag{6}
$$

Second, to improve the transferability of $\tilde{x}$, we expect that $\tilde{x}^*$ remains effective to $f_s(\cdot)$, so the second part of $\mathcal{L}_G$ can be derived as:

$$
\begin{aligned}
\mathcal{L}_{G2}(f_s(\tilde{x}^*), C_{tr}) &= -\log\left(\frac{\sum_{i \neq C_{tr}} \exp(f_s(\tilde{x}^*)_i)}{\sum_i \exp(f_s(\tilde{x}^*)_i)}\right) \\
&= -\log\left(1 - \frac{\exp(f_s(\tilde{x}^*)_{C_{tr}})}{\sum_i \exp(f_s(\tilde{x}^*)_i)}\right)
\end{aligned}
\tag{7}
$$

Finally, the last part of $\mathcal{L}_G$ is used to limit the perturbation magnitude. We introduce the traditional $L_p$-norm to measure the degree of image distortion as follows:

$$
\begin{aligned}
\mathcal{L}_{G3}(x,\tilde{x}) &= \|\tilde{x} - x\|_p \\
&= \left(\sum_i |\Delta x_i|^p\right)^{\frac{1}{p}}
\end{aligned} \tag{8}
$$

In summary, we apply the linear weighted sum method to balance the relationship among $\mathcal{L}_{G1}$, $\mathcal{L}_{G2}$, and $\mathcal{L}_{G3}$. As such, the complete generator loss for the non-targeted attack can be represented as:

$$
\mathcal{L}_G = \omega_{G1} \cdot \mathcal{L}_{G1}(f_s(\tilde{x}), C_{tr}) + \omega_{G2} \cdot \mathcal{L}_{G2}(f_s(\tilde{x}^*), C_{tr}) + \omega_{G3} \cdot \mathcal{L}_{G3}(x, \tilde{x}) \tag{9}
$$

where $\omega_{G1} + \omega_{G2} + \omega_{G3} = 1$. $\omega_{G1}, \omega_{G2}, \omega_{G3} \in [0,1]$ are the weight coefficients of $\mathcal{L}_{G1}$, $\mathcal{L}_{G2}$, and $\mathcal{L}_{G3}$, respectively. The weight coefficients represent the relative importance of each loss term during the training process. A larger weight implies that the corresponding loss will decrease more rapidly and significantly, allowing attackers to adjust the parameters flexibly according to their actual needs.

For the targeted attack: According to (3), $\tilde{x}$ is to maximize the probability of being recognized as the target class $C_{ta}$, i.e., to increase the confidence of $C_{ta}$. Thus, $\mathcal{L}_{G1}$ here can be expressed as:

$$
\mathcal{L}_{G1}(f_s(\tilde{x}), C_{ta}) = -\log\left(\frac{\exp(f_s(\tilde{x})_{C_{ta}})}{\sum_i \exp(f_s(\tilde{x})_i)}\right) \tag{10}
$$

To maintain the effectiveness of $\tilde{x}^*$ against $f_s(\cdot)$, $\mathcal{L}_{G2}$ here is derived as:

$$
\mathcal{L}_{G2}(f_s(\tilde{x}^*), C_{ta}) = -\log\left(\frac{\exp(f_s(\tilde{x}^*)_{C_{ta}})}{\sum_i \exp(f_s(\tilde{x}^*)_i)}\right) \tag{11}
$$

The perturbation magnitude is still limited by the $\mathcal{L}_{G3}$ shown in (8). Therefore, the complete generator loss for the targeted attack can be represented as:

$$
\mathcal{L}_G = \omega_{G1} \cdot \mathcal{L}_{G1}(f_s(\tilde{x}), C_{ta}) + \omega_{G2} \cdot \mathcal{L}_{G2}(f_s(\tilde{x}^*), C_{ta}) + \omega_{G3} \cdot \mathcal{L}_{G3}(x, \tilde{x}) \tag{12}
$$

*3.2. Training Process of the Attenuator*

According to Figure 2, during the training phase of TAN, an attenuator $A(\cdot)$ was introduced to weaken the effectiveness of $\tilde{x}$ while still preserving the semantic meaning of $x$. We show the detailed training process of $A(\cdot)$ in Figure 4.

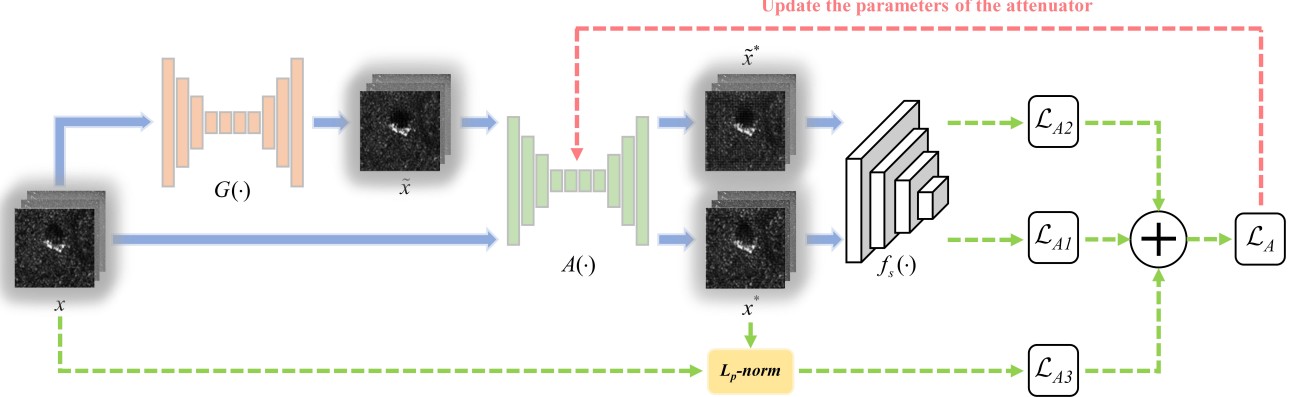

**Figure 4.** Training process of the attenuator.

As we can see, the attenuator loss $\mathcal{L}_A$ also consists of three parts. First, to preserve the semantic meaning of $x$, $f_s(\cdot)$ has to keep a basic classification accuracy on the following attenuated sample $x^*$:

$$x^* = A(x) \tag{13}$$

It means that the first part of $\mathcal{L}_A$ should increase the confidence of $x^*$ being recognized as the true class $C_{tr}$, as follows:

$$\mathcal{L}_{A1}(f_s(x^*), C_{tr}) = -\log\left(\frac{\exp(f_s(x^*)_{C_{tr}})}{\sum_i \exp(f_s(x^*)_i)}\right) \tag{14}$$

Meanwhile, to weaken the effectiveness of $\tilde{x}$, $A(\cdot)$ also need to improve the confidence of the attenuated adversarial example $\tilde{x}^*$ being identified as $C_{tr}$, so the second part of $\mathcal{L}_A$ can be expressed as:

$$\mathcal{L}_{A2}(f_s(\tilde{x}^*), C_{tr}) = -\log\left(\frac{\exp(f_s(\tilde{x}^*)_{C_{tr}})}{\sum_i \exp(f_s(\tilde{x}^*)_i)}\right) \tag{15}$$

Finally, to avoid excessive image distortion caused by $A(\cdot)$, the third part of $\mathcal{L}_A$ is used to limit the deformation magnitude, which can be expressed by the traditional $L_p$-norm, as follows:

$$\begin{aligned} \mathcal{L}_{A3}(x, x^*) &= \|x^* - x\|_p \\ &= \left(\sum_i |\Delta x_i|^p\right)^{\frac{1}{p}} \end{aligned} \tag{16}$$

As with the generator loss, we utilize the linear weighted sum method to derive the complete attenuator loss as follows:

$$\mathcal{L}_A = \omega_{A1} \cdot \mathcal{L}_{A1}(f_s(x^*), C_{tr}) + \omega_{A2} \cdot \mathcal{L}_{A2}(f_s(\tilde{x}^*), C_{tr}) + \omega_{A3} \cdot \mathcal{L}_{A3}(x, x^*) \tag{17}$$

where $\omega_{A1} + \omega_{A2} + \omega_{A3} = 1$. $\omega_{A1}, \omega_{A2}, \omega_{A3} \in [0, 1]$ are the weight coefficients of $\mathcal{L}_{A1}, \mathcal{L}_{A2}$, and $\mathcal{L}_{A3}$, respectively.

### 3.3. Network Structure of the Generator and Attenuator

According to Sections 3.1 and 3.2, the generator and attenuator are essentially two encoder–decoder models, so the choice of a suitable model structure is necessary. We mainly consider two factors. First, as the size of original samples and adversarial examples should be the same, the model has to keep the input and output sizes identical. Second, to prevent our network from overfitting while saving computational resources, a lightweight model will be a better choice. In summary, we applied ResNet Generator proposed in [29] as the encoder–decoder model of TAN. The structure of ResNet Generator is shown in Figure 5.

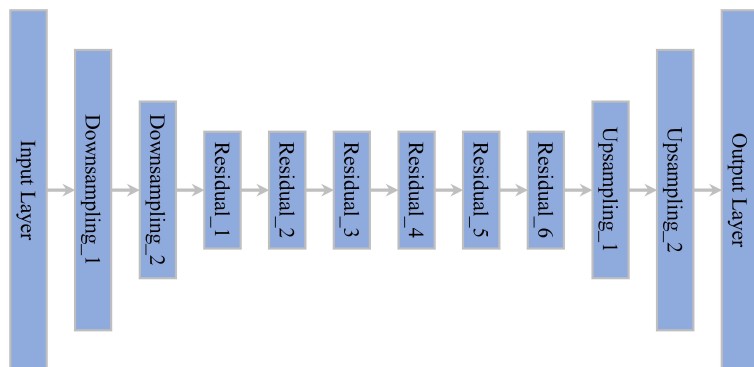

**Figure 5.** Structure of ResNet Generator.

As we can see, ResNet Generator mainly consists of downsampling, residual, and upsampling modules. For a visual understanding, given an input data of size $1 \times 128 \times 128$, the input and output sizes of each module are listed in Table 1.

Obviously, the input and output sizes of ResNet Generator are the same. Meanwhile, to ensure the validity of the generated data, we added a *tanh* function after the output module, which restricts the generated data to the interval $[0, 1]$. The total number of parameters in ResNet Generator has been calculated to be approximately $7.83M$, which is a fairly lightweight network. For more details, please refer to the literature [29].

**Table 1.** Input–output relationships for each module of ResNet Generator.

| Module | Input Size | Output Size |
|---|---|---|
| Input | $1 \times 128 \times 128$ | $64 \times 128 \times 128$ |
| Downsampling_1 | $64 \times 128 \times 128$ | $128 \times 64 \times 64$ |
| Downsampling_2 | $128 \times 64 \times 64$ | $256 \times 32 \times 32$ |
| Residual_1 $\sim$ 6 | $256 \times 32 \times 32$ | $256 \times 32 \times 32$ |
| Upsampling_1 | $256 \times 32 \times 32$ | $128 \times 64 \times 64$ |
| Upsampling_2 | $128 \times 64 \times 64$ | $64 \times 128 \times 128$ |
| Output | $64 \times 128 \times 128$ | $1 \times 128 \times 128$ |

### 3.4. Complete Training Process of TAN

As we described earlier, TAN improves the transferability of adversarial examples through a two-player game between the generator and attenuator, which is quite similar to the working principle of generative adversarial networks (GAN) [30]. Therefore, we also adopted an alternating training scheme to train our network. Specifically, given the dataset $\mathcal{X}$ and batch size $S$, we first randomly divided $\mathcal{X}$ into $M$ batches $\{b_1, b_2, \cdots, b_M\}$ at the beginning of each training iteration. Second, we set a training ratio $R \in N^*$, which means that TAN trains the generator $R$ times and then trains the attenuator once, i.e., once per batch for the former and only once per $R$ batch for the latter. In this way, we can prevent the attenuator from being so strong that the generator cannot be optimized. Meanwhile, to shorten training time, we set an early stop condition *Esc* so that training can be ended early when certain indicators meet the condition. Note that the generator and attenuator are trained alternately, i.e., the attenuator's parameters are fixed when the generator is trained, and vice versa. More details of the complete training process for TAN are shown in Algorithm 1.

## 4. Experiments

### 4.1. Data Descriptions

To date, there is no publicly available dataset for UAV SAR-ATR, thus this paper experiments on the most authoritative SAR-ATR dataset, i.e., the moving and stationary target acquisition and recognition (MSTAR) dataset [31]. MSTAR is collected by a high-resolution spotlight SAR and published by the U.S. Defense Advanced Research Projects Agency (DARPA) in 1996, which contains SAR images of Soviet military vehicle targets at different azimuth and depression angles. In standard operating conditions (SOC), MSTAR includes ten classes of targets, such as self-propelled howitzers (2S1); infantry fighting vehicles (BMP2); armored reconnaissance vehicles (BRDM2); wheeled armored transport vehicles (BTR60, BTR70); bulldozers (D7); main battle tanks (T62, T72); cargo trucks (ZIL131); and self-propelled artillery (ZSU234). The training dataset contains 2747 images collected at a depression angle of $17°$, and the testing dataset contains 2426 images captured at a depression angle of $15°$. More details about the dataset are given in Table 2, and Figure 6 shows the optical images and corresponding SAR images of each class.

---

**Algorithm 1** Transferable Adversarial Network Training

---

**Input:** Dataset $\mathcal{X}$; batch size $S$; surrogate model $f_s$; target class $C_{ta}$; training loss function $\mathcal{L}_G$ of the generator; training loss function $\mathcal{L}_A$ of the attenuator; training iteration number $T$; learning rate $\eta$; training ratio $R$ of the generator and attenuator; early stop condition *Esc*.

**Output:** The parameter $\theta_G$ of the well-trained generator.

1: Randomly initialize $\theta_G$ and $\theta_A$
2: **for** $t = 1$ to $T$ **do**
3:     According to $S$, randomly divide $\mathcal{X}$ into $M$ batches $\{b_1, b_2, \cdots, b_M\}$
4:     **for** $m = 1$ to $M$ **do**
5:         Calculate $\mathcal{L}_G(\theta_G, \theta_A, f_s, b_m, C_{ta})$
6:         Update $\theta_G = \theta_G - \eta \cdot \frac{\partial}{\partial \theta_G} \mathcal{L}_G$
7:         **if** $m \% R == 0$ **then**
8:             Calculate $\mathcal{L}_A(\theta_G, \theta_A, f_s, b_m)$
9:             Update $\theta_A = \theta_A - \eta \cdot \frac{\partial}{\partial \theta_A} \mathcal{L}_A$
10:         **else**
11:             $\theta_A = \theta_A$
12:         **end if**
13:     **end for**
14:     **if** $Esc == True$ **then**
15:         Break
16:     **else**
17:         Continue
18:     **end if**
19: **end for**

---

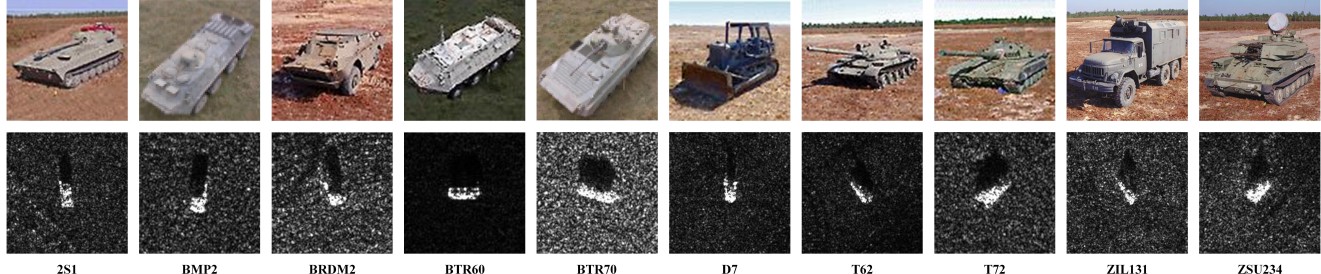

**Figure 6.** Optical images (**top**) and SAR images (**bottom**) of the MSTAR dataset.

**Table 2.** Details of the MSTAR dataset under SOC, including target class, serial, depression angle, and sample numbers.

| Target Class | Serial | Training Data | | Testing Data | |
|---|---|---|---|---|---|
| | | Depression Angle | Number | Depression Angle | Number |
| 2S1 | b01 | 17° | 299 | 15° | 274 |
| BMP2 | 9566 | 17° | 233 | 15° | 196 |
| BRDM2 | E-71 | 17° | 298 | 15° | 274 |
| BTR60 | k10yt7532 | 17° | 256 | 15° | 195 |
| BTR70 | c71 | 17° | 233 | 15° | 196 |
| D7 | 92v13015 | 17° | 299 | 15° | 274 |
| T62 | A51 | 17° | 299 | 15° | 273 |
| T72 | 132 | 17° | 232 | 15° | 196 |
| ZIL131 | E12 | 17° | 299 | 15° | 274 |
| ZSU234 | d08 | 17° | 299 | 15° | 274 |

### 4.2. Implementation Details

The proposed method is evaluated on the following six common DNN models: DenseNet121 [32], GoogLeNet [33], InceptionV3 [34], Mobilenet [35], ResNet50 [36], and Shufflenet [37]. In terms of data preprocessing, we resized all the images in MSTAR to $128 \times 128$ and uniformly sample 10% of training data to form the validation dataset. During the training phase of recognition models, the training epoch and batch size were set to 100 and 32, respectively. During the training phase of TAN, to minimize the MSE between adversarial examples and original samples, we adopted the $L_2$-norm to evaluate the image distortion caused by adversarial perturbations. Meanwhile, for better attack performance, the hyperparameters of TAN are fine-tuned through numerous experiments, and the following set of parameters is eventually determined to best meet our requirements. Specifically, we set the generator loss weights $[\omega_{G1}, \omega_{G2}, \omega_{G3}]$ to $[0.25, 0.25, 0.5]$, the attenuator loss weights $[\omega_{A1}, \omega_{A2}, \omega_{A3}]$ to $[0.25, 0.25, 0.5]$, the training ratio to 3, the training epoch to 50, and the batch size to 8. Due to the adversarial process involved in TAN, training can be challenging to converge. As such, we employed Adam [38], a more computationally efficient optimizer, to accelerate model convergence, which also performs better in solving non-stationary objective and sparse gradient problems. The learning rate is set to 0.001. When evaluating the transferability, we first crafted adversarial examples for each surrogate model and then assessed the transferability by testing the recognition results of victim models on corresponding adversarial examples. Detailed experiments will be given later.

Furthermore, the following six attack algorithms from the Torchattacks [39] toolbox were introduced as baseline methods for comparison with TAN: MIFGSM [19], DIFGSM [21], NIFGSM [20], SINIFGSM [20], VMIFGSM [22], and VNIFGSM [22]. All codes were written in Pytorch, and the experimental environment consisted of Windows 10 with an NVIDIA GeForce RTX 2080 Ti GPU and a 3.6 GHz Intel Core i9-9900K CPU.

### 4.3. Evaluation Metrics

We mainly consider two factors to comprehensively evaluate the performance of adversarial attacks: the effectiveness and stealthiness, which are directly related to the classification accuracy $\tilde{Acc}$ of victim models on adversarial examples and the norm value $\tilde{L}_p$ of adversarial perturbations, respectively. For the $\tilde{Acc}$ metric, the formula is as follows:

$$\tilde{Acc} = \begin{cases} \frac{1}{N} \sum_{n=1}^{N} D(\arg\max_i (f_v(\tilde{x}_n)_i) == C_{tr}) & \text{for the non-targeted attack} \\[2ex] \frac{1}{K \times N} \sum_{C_{ta}=1}^{K} \sum_{n=1}^{N} D(\arg\max_i (f_v(\tilde{x}_n)_i) == C_{ta}) & \text{for the targeted attack} \end{cases} \quad (18)$$

where $C_{tr}$ and $C_{ta}$ represent the true and target classes of the input data, $K$ is the number of target classes, and $D(\cdot)$ is a discriminant function. In the non-targeted attack, the $\tilde{Acc}$ metric reflects the probability that the victim model $f_v(\cdot)$ identifies the adversarial example $\tilde{x}_n$ as $C_{tr}$, while in the targeted attack it indicates the probability that $f_v(\cdot)$ recognizes $\tilde{x}_n$ as $C_{ta}$. Obviously, in the non-targeted attack, the lower the $\tilde{Acc}$ metric, the better the attack. Conversely, in the targeted attack, a higher $\tilde{Acc}$ metric represents $f_v(\cdot)$ is more likely to recognize $\tilde{x}_n$ as $C_{ta}$, and thus the attack is more effective. In conclusion, the effectiveness of non-targeted attacks is inversely proportional to the $\tilde{Acc}$ metric, and the effectiveness of targeted attacks is proportional to this metric. Additionally, there are other three similar indicators, $Acc$, $Acc^*$, and $\tilde{Acc}^*$, that represent the classification accuracy of $f_v(\cdot)$ for the original sample $x_n$, the attenuated sample $x_n^*$, and the attenuated adversarial example $\tilde{x}_n^*$, respectively. Note that whether it is a non-targeted or targeted attack, $Acc^*$ always represents the accuracy with which $f_v(\cdot)$ identifies $x_n^*$ as $C_{tr}$, while the other three accuracy indicators need to be calculated via (18) based on the attack mode. In particular, $\tilde{Acc}^*$ represents the recognition result of $f_v(\cdot)$ on $\tilde{x}_n^*$, which indirectly reflects the strength of the transferability possessed by $\tilde{x}_n$.

Meanwhile, we applied the following $L_p$-norm values to measure the attack stealthiness:

$$
\begin{cases}
\tilde{L}_p = \frac{1}{N} \sum_{n=1}^{N} \|\tilde{x}_n - x_n\|_p & \text{for the generator} \\[3mm]
L_p^* = \frac{1}{N} \sum_{n=1}^{N} \|x_n^* - x_n\|_p & \text{for the attenuator}
\end{cases}
\tag{19}
$$

where $\tilde{L}_p$ and $L_p^*$ represent the image distortion caused by the generator and attenuator, respectively. In our experiments, the $L_p$-norm defaults to $L_2$-norm. In summary, we can set the early stop condition $Esc$ mentioned in Section 3.4 with the above indicators, as follows:

$$
Esc = \begin{cases}
\tilde{Acc} \leq 0.05, Acc^* \geq 0.9, \tilde{Acc}^* \leq 0.1, \tilde{L}_2 \leq 4, L_2^* \leq 4 & \text{for the non-targeted attack} \\[3mm]
\tilde{Acc} \geq 0.95, Acc^* \geq 0.9, \tilde{Acc}^* \geq 0.9, \tilde{L}_2 \leq 4, L_2^* \leq 4 & \text{for the targeted attack}
\end{cases}
\tag{20}
$$

Furthermore, to evaluate the real-time performance of adversarial attacks, we introduced the $Tc$ metric to denote the time cost of generating a single adversarial example, as follows:

$$
Tc = \frac{Time}{N}
\tag{21}
$$

where $Time$ is the total time consumed to generate $N$ adversarial examples.

### 4.4. DNN-Based SAR-ATR Models

A well-trained recognition model is a prerequisite for effective adversarial attacks, so we have trained six SAR-ATR models on the MSTAR dataset: DenseNet121, GoogLeNet, InceptionV3, Mobilenet, ResNet50, and Shufflenet. All of them achieve outstanding recognition performance, with the classification accuracy of 98.72%, 98.06%, 96.17%, 96.91%, 97.98%, and 96.66% on the testing dataset, respectively. In addition, we show the confusion matrix of each model in Figure 7.

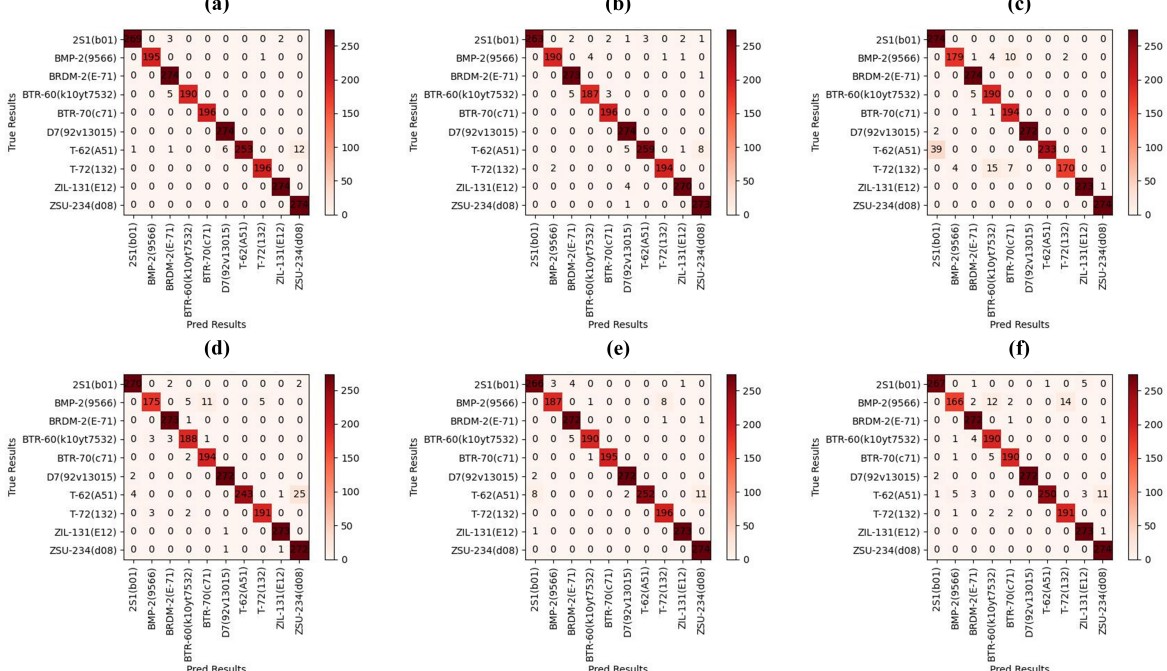

**Figure 7.** Confusion matrixes of DNN-based SAR-ATR models on the MSTAR dataset. (**a**) DenseNet121. (**b**) GoogLeNet. (**c**) InceptionV3. (**d**) Mobilenet. (**e**) ResNet50. (**f**) Shufflenet.

### 4.5. Comparison of Attack Performance

In this section, we first evaluated the attack performance of the proposed method against DNN-based SAR-ATR models on the MSTAR dataset. Specifically, during the training phase of TAN, we took each network as the surrogate model in turn and assessed the recognition results of corresponding models on the outputs of TAN at each stage. The results of non-targeted and targeted attacks are detailed in Tables 3 and 4, respectively.

**Table 3.** Non-targeted attack results of our method against DNN-based SAR-ATR models on the MSTAR dataset.

| Surrogate | $Acc$ | $\tilde{Acc}$ | $Acc^*$ | $\tilde{Acc}^*$ | $\tilde{L}_2$ | $L_2^*$ |
|---|---|---|---|---|---|---|
| DenseNet121 | 98.72% | 1.90% | 81.53% | 24.03% | 3.595 | 4.959 |
| GoogLeNet | 98.06% | 3.83% | 89.78% | 36.11% | 2.884 | 3.305 |
| InceptionV3 | 96.17% | 0.82% | 89.41% | 19.62% | 3.552 | 4.181 |
| Mobilenet | 96.91% | 2.72% | 87.88% | 36.81% | 3.218 | 4.083 |
| ResNet50 | 97.98% | 3.34% | 83.80% | 28.65% | 3.684 | 4.568 |
| Shufflenet | 96.66% | 3.46% | 84.30% | 23.66% | 3.331 | 3.286 |
| Mean | 97.42% | 2.68% | 86.12% | 28.15% | 3.377 | 4.064 |

**Table 4.** Targeted attack results of our method against DNN-based SAR-ATR models on the MSTAR dataset.

| Surrogate | $Acc$ | $\tilde{Acc}$ | $Acc^*$ | $\tilde{Acc}^*$ | $\tilde{L}_2$ | $L_2^*$ |
|---|---|---|---|---|---|---|
| DenseNet121 | 10.00% | 98.08% | 88.47% | 78.09% | 3.086 | 3.587 |
| GoogLeNet | 10.00% | 99.09% | 89.25% | 85.90% | 3.377 | 4.289 |
| InceptionV3 | 10.00% | 98.81% | 86.87% | 78.97% | 3.453 | 3.495 |
| Mobilenet | 10.00% | 97.40% | 88.38% | 81.37% | 3.257 | 3.553 |
| ResNet50 | 10.00% | 97.69% | 87.29% | 82.10% | 3.408 | 3.490 |
| Shufflenet | 10.00% | 98.36% | 86.85% | 83.11% | 3.345 | 3.874 |
| Mean | 10.00% | 98.24% | 87.85% | 81.59% | 3.321 | 3.714 |

In non-targeted attacks, the $Acc$ metric of each model on the MSTAR dataset exceeds 95%. However, after the non-targeted attack, the classification accuracy of all models on the generated adversarial examples, i.e., the $\tilde{Acc}$ metric, is below 5%, and the $\tilde{L}_2$ indicator is less than 3.7. It means that adversarial examples deteriorate the recognition performance of models rapidly through minor adversarial perturbations. Meanwhile, during the training phase of TAN, we evaluate the performance of the attenuator. According to the $\tilde{Acc}^*$ metric, the attenuator leads to an average improvement of about 25% in the classification accuracy of models on adversarial examples, that is, it indeed weakens the effectiveness of adversarial examples. We also should pay attention to the metrics $Acc^*$ and $L_2^*$, i.e., the recognition accuracy of models on the attenuated samples, and the deformation distortion caused by the attenuator. The fact is that the $Acc^*$ indicator of each model exceeds 80%, and the average value of the $L_2^*$ metric is about 4. It means that the attenuator retains most semantic information of original samples without causing excessive deformation distortion, which is in line with our requirements.

In targeted attacks, the $Acc$ metric represents the probability that models identify original samples as target classes, so it can reflect the dataset distribution, i.e., each category accounts for about 10% of the total dataset. After the targeted attack, the probability of each model recognizing adversarial examples as target classes, i.e., the $\tilde{Acc}$ metric, is over 97%, and the $\tilde{L}_2$ indicator shows that the image distortion caused by adversarial perturbations is less than 3.5. It means that the adversarial examples crafted by the generator can induce models to output specified results with high probability through minor perturbations. As with the non-targeted attack, we evaluate the performance of the attenuator. The $\tilde{Acc}^*$ metric shows that the attenuator results in an average decrease of about 17% in the probability of adversarial examples being identified as target classes. Meanwhile, the $Acc^*$

metric of each model exceeds 85%, and the average value of the $L_2^*$ indicator is about 3.7. That is, the attenuator weakens the effectiveness of adversarial examples through slight deformations, while preserving the semantic meaning of original samples well.

In summary, for both non-targeted and targeted attacks, the adversarial examples crafted by the generator can fool models with high success rates, and the attenuator is able to weaken the effectiveness of adversarial examples with slight deformations while retaining the semantic meaning of original samples. Moreover, we ensure that the generator always outperforms the attenuator by adjusting the training ratio between the two models. To visualize the attack results of TAN, we took ResNet50 as the surrogate model and display the outputs of TAN at each stage in Figure 8.

Finally, we compared the non-targeted and targeted attack performance of different methods against DNN-based SAR-ATR models on the MSTAR dataset, as detailed in Table 5. Obviously, for the same image distortion, the attack effectiveness of the proposed method against a single model may not be the best. Nevertheless, we focused more on the transferability of adversarial examples, which will be the main topic of the following section.

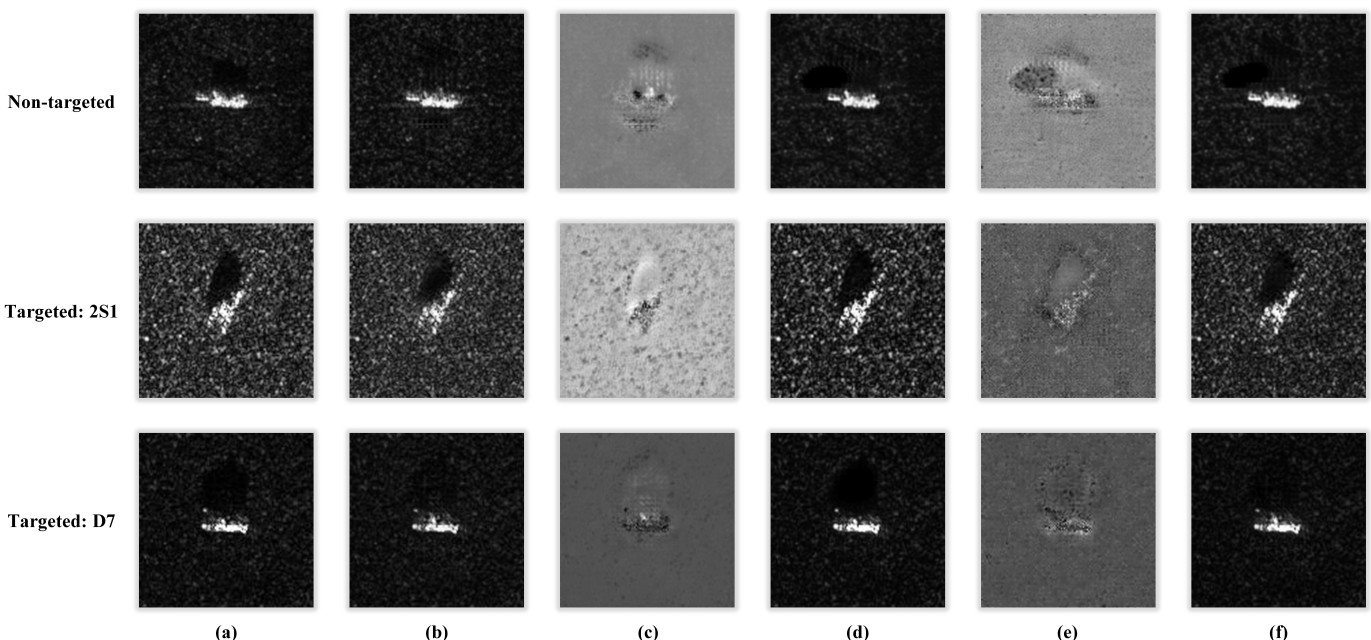

**Figure 8.** Visualization of attack results against ResNet50. (**a**) Original samples. (**b**) Adversarial examples. (**c**) Adversarial perturbations. (**d**) Attenuated samples. (**e**) Deformation distortion. (**f**) Attenuated adversarial examples. From top to bottom, the corresponding target classes are None, 2S1, and D7, respectively.

**Table 5.** Attack performance of different methods against DNN-based SAR-ATR models on the MSTAR dataset.

| Surrogate | Method | Non-Targeted | | Targeted | |
|---|---|---|---|---|---|
| | | $\tilde{Acc}$ | $\tilde{L}_2$ | $\tilde{Acc}$ | $\tilde{L}_2$ |
| DenseNet121 | TAN | 1.90% | 3.595 | 98.08% | 3.086 |
| | MIFGSM | 0.00% | 3.555 | 98.61% | 3.613 |
| | DIFGSM | 0.00% | 3.116 | 95.39% | 2.816 |
| | NIFGSM | 0.21% | 3.719 | 68.72% | 3.550 |
| | SINIFGSM | 1.15% | 3.676 | 82.32% | 3.648 |
| | VMIFGSM | 0.00% | 3.665 | 98.14% | 3.602 |
| | VNIFGSM | 0.08% | 3.691 | 96.89% | 3.635 |

**Table 5.** *Cont.*

| Surrogate | Method | Non-Targeted | | Targeted | |
|---|---|---|---|---|---|
| | | $\tilde{Acc}$ | $\tilde{L}_2$ | $\tilde{Acc}$ | $\tilde{L}_2$ |
| GoogLeNet | TAN | 3.83% | 2.884 | 99.09% | 3.377 |
| | MIFGSM | 0.04% | 3.615 | 98.36% | 3.601 |
| | DIFGSM | 0.04% | 3.090 | 94.47% | 2.830 |
| | NIFGSM | 0.41% | 3.674 | 64.32% | 3.520 |
| | SINIFGSM | 4.04% | 3.647 | 69.79% | 3.615 |
| | VMIFGSM | 0.04% | 3.587 | 97.84% | 3.601 |
| | VNIFGSM | 0.37% | 3.588 | 95.74% | 3.636 |
| InceptionV3 | TAN | 0.82% | 3.552 | 98.81% | 3.453 |
| | MIFGSM | 0.00% | 3.599 | 96.00% | 3.563 |
| | DIFGSM | 0.04% | 3.010 | 86.72% | 2.811 |
| | NIFGSM | 0.21% | 3.671 | 51.66% | 3.397 |
| | SINIFGSM | 2.93% | 3.689 | 62.46% | 3.593 |
| | VMIFGSM | 0.00% | 3.614 | 91.54% | 3.577 |
| | VNIFGSM | 0.00% | 3.632 | 84.02% | 3.605 |
| Mobilenet | TAN | 2.72% | 3.218 | 97.40% | 3.257 |
| | MIFGSM | 8.29% | 3.557 | 99.86% | 3.538 |
| | DIFGSM | 6.64% | 2.821 | 91.64% | 2.610 |
| | NIFGSM | 6.88% | 3.575 | 80.05% | 3.519 |
| | SINIFGSM | 1.77% | 3.664 | 85.14% | 3.662 |
| | VMIFGSM | 2.35% | 3.572 | 99.40% | 3.499 |
| | VNIFGSM | 1.32% | 3.635 | 95.58% | 3.582 |
| ResNet50 | TAN | 3.34% | 3.684 | 97.69% | 3.408 |
| | MIFGSM | 0.95% | 3.659 | 97.08% | 3.613 |
| | DIFGSM | 0.33% | 3.141 | 90.35% | 2.824 |
| | NIFGSM | 0.33% | 3.710 | 45.34% | 3.501 |
| | SINIFGSM | 3.96% | 3.720 | 71.64% | 3.652 |
| | VMIFGSM | 0.87% | 3.644 | 96.17% | 3.618 |
| | VNIFGSM | 0.25% | 3.692 | 94.17% | 3.632 |
| Shufflenet | TAN | 3.46% | 3.331 | 98.36% | 3.345 |
| | MIFGSM | 0.00% | 3.567 | 100.00% | 3.518 |
| | DIFGSM | 0.00% | 2.790 | 97.54% | 2.599 |
| | NIFGSM | 0.16% | 3.632 | 91.77% | 3.455 |
| | SINIFGSM | 0.00% | 3.660 | 95.79% | 3.568 |
| | VMIFGSM | 0.00% | 3.617 | 100.00% | 3.511 |
| | VNIFGSM | 0.04% | 3.654 | 99.73% | 3.568 |

*4.6. Comparison of Transferability*

In this section, we evaluated the transferability of adversarial examples among DNN-based SAR-ATR models on the MSTAR dataset. Specifically, we first took each network as the surrogate model in turn and crafted adversarial examples for them, respectively. Then, we assessed the transferability by testing the recognition results of victim models on corresponding adversarial examples. The transferability in non-targeted and targeted attacks are shown in Tables 6 and 7, respectively.

In non-targeted attacks, when the proposed method sequentially takes DenseNet121, GoogLeNet, InceptionV3, Mobilenet, ResNet50, and Shufflenet as the surrogate model, the highest recognition accuracy of victim models on the generated adversarial examples are 12.90%, 26.88%, 23.45%, 18.59%, 11.01%, and 23.54%, respectively. Equivalently, the highest recognition accuracy of victim models on the adversarial examples produced by baseline methods are 36.11%, 44.44%, 56.06%, 65.99%, 33.84%, and 68.51%, respectively. Meanwhile, for each surrogate model, victim models always have the lowest recognition accuracy on the adversarial examples crafted by our approach. Obviously, compared with baseline methods, the proposed method slightly sacrifices the performance on attacking surrogate

models, but achieves state-of-the-art transferability among victim models in non-targeted attacks. Detailed results are shown in Table 6.

**Table 6.** Transferability of adversarial examples generated by different attack algorithms in non-targeted attacks.

| Surrogate | Method | DenseNet121 | GoogLeNet | InceptionV3 | Mobilenet | ResNet50 | Shufflenet |
|-----------|--------|-------------|-----------|-------------|-----------|----------|------------|
| DenseNet121 | TAN | 1.90% | 4.25% | 7.46% | 9.93% | 9.11% | 12.90% |
| | MIFGSM | 0.00% | 10.10% | 12.82% | 26.46% | 16.32% | 28.65% |
| | DIFGSM | 0.00% | 8.16% | 11.46% | 26.01% | 19.17% | 30.83% |
| | NIFGSM | 0.21% | 14.67% | 14.67% | 26.75% | 20.07% | 30.67% |
| | SINIFGSM | 1.15% | 16.69% | 19.29% | 35.66% | 17.64% | 36.11% |
| | VMIFGSM | 0.00% | 8.86% | 11.62% | 24.40% | 15.13% | 25.89% |
| | VNIFGSM | 0.08% | 8.04% | 11.62% | 22.38% | 13.60% | 23.54% |
| GoogLeNet | TAN | 6.88% | 3.83% | 8.16% | 23.62% | 10.51% | 26.88% |
| | MIFGSM | 10.18% | 0.04% | 17.72% | 32.36% | 27.66% | 42.13% |
| | DIFGSM | 8.33% | 0.04% | 14.47% | 32.52% | 24.73% | 38.66% |
| | NIFGSM | 22.88% | 0.41% | 24.28% | 32.32% | 35.16% | 44.44% |
| | SINIFGSM | 7.96% | 4.04% | 13.15% | 33.22% | 15.09% | 28.07% |
| | VMIFGSM | 8.57% | 0.04% | 16.32% | 29.72% | 25.64% | 38.58% |
| | VNIFGSM | 10.02% | 0.37% | 15.50% | 27.99% | 26.30% | 36.93% |
| InceptionV3 | TAN | 8.20% | 9.60% | 0.82% | 21.43% | 14.67% | 23.45% |
| | MIFGSM | 19.25% | 35.00% | 0.00% | 39.45% | 33.14% | 42.54% |
| | DIFGSM | 16.86% | 33.22% | 0.04% | 43.69% | 33.76% | 47.07% |
| | NIFGSM | 32.11% | 34.46% | 0.21% | 42.09% | 43.08% | 44.89% |
| | SINIFGSM | 27.37% | 38.05% | 2.93% | 49.22% | 41.18% | 56.06% |
| | VMIFGSM | 18.51% | 26.92% | 0.00% | 34.46% | 31.04% | 37.18% |
| | VNIFGSM | 21.68% | 26.38% | 0.00% | 33.80% | 34.50% | 37.63% |
| Mobilenet | TAN | 14.34% | 15.83% | 13.56% | 2.72% | 14.18% | 18.59% |
| | MIFGSM | 65.99% | 59.32% | 53.59% | 8.29% | 55.56% | 59.77% |
| | DIFGSM | 51.28% | 53.34% | 49.34% | 6.64% | 49.34% | 52.18% |
| | NIFGSM | 65.75% | 58.66% | 51.85% | 6.88% | 52.31% | 55.56% |
| | SINIFGSM | 64.67% | 45.14% | 49.01% | 1.77% | 51.81% | 58.37% |
| | VMIFGSM | 62.49% | 52.10% | 50.45% | 2.35% | 49.63% | 52.84% |
| | VNIFGSM | 56.27% | 50.04% | 43.61% | 1.32% | 43.82% | 48.19% |
| ResNet50 | TAN | 5.94% | 9.27% | 10.14% | 12.94% | 3.34% | 11.01% |
| | MIFGSM | 14.59% | 24.15% | 17.72% | 16.90% | 0.95% | 26.42% |
| | DIFGSM | 11.13% | 17.07% | 15.09% | 20.45% | 0.33% | 26.59% |
| | NIFGSM | 21.72% | 28.19% | 20.28% | 19.74% | 0.33% | 29.43% |
| | SINIFGSM | 26.50% | 24.15% | 22.59% | 30.50% | 3.96% | 33.84% |
| | VMIFGSM | 13.31% | 22.42% | 16.36% | 15.95% | 0.87% | 23.33% |
| | VNIFGSM | 15.00% | 22.67% | 16.45% | 14.47% | 0.25% | 22.63% |
| Shufflenet | TAN | 17.72% | 23.54% | 16.49% | 22.22% | 17.85% | 3.46% |
| | MIFGSM | 66.69% | 70.03% | 65.00% | 55.81% | 65.00% | 0.00% |
| | DIFGSM | 53.46% | 57.58% | 55.32% | 51.44% | 55.44% | 0.00% |
| | NIFGSM | 67.23% | 61.58% | 58.62% | 48.35% | 61.62% | 0.16% |
| | SINIFGSM | 68.51% | 58.33% | 60.92% | 50.41% | 56.64% | 0.00% |
| | VMIFGSM | 57.25% | 55.32% | 54.29% | 40.23% | 53.34% | 0.00% |
| | VNIFGSM | 56.68% | 54.25% | 51.57% | 37.30% | 52.14% | 0.04% |

In targeted attacks, the proposed method still takes DenseNet121, GoogLeNet, InceptionV3, Mobilenet, ResNet50, and Shufflenet as the surrogate model in turn, and the minimum probability that victim models identify the generated adversarial examples as target classes are 52.39%, 55.02%, 54.57%, 57.66%, 66.26%, and 47.78%, respectively. Correspondingly, the minimum probability that victim models recognize the adversarial examples produced by baseline methods as target classes are 22.18%, 19.63%, 19.49%, 15.52%, 19.36%, and 13.06%, respectively. Moreover, for each surrogate model, victim

models always identify the adversarial examples crafted by our approach as target classes with the maximum probability. Thus, the proposed method also achieves state-of-the-art transferability among victim models in targeted attacks. Detailed results are shown in Table 7.

**Table 7.** Transferability of adversarial examples generated by different attack algorithms in targeted attacks.

| Surrogate | Method | DenseNet121 | GoogLeNet | InceptionV3 | Mobilenet | ResNet50 | Shufflenet |
|---|---|---|---|---|---|---|---|
| DenseNet121 | TAN | 98.08% | 79.12% | 70.71% | 59.03% | 62.31% | 52.39% |
| | MIFGSM | 98.61% | 52.47% | 49.05% | 39.47% | 43.78% | 37.62% |
| | DIFGSM | 95.39% | 51.08% | 46.62% | 35.02% | 39.51% | 32.29% |
| | NIFGSM | 68.72% | 33.06% | 27.61% | 22.18% | 25.78% | 22.92% |
| | SINIFGSM | 82.32% | 40.62% | 33.17% | 29.95% | 31.93% | 30.59% |
| | VMIFGSM | 98.14% | 48.94% | 44.10% | 33.56% | 39.29% | 34.06% |
| | VNIFGSM | 96.89% | 48.78% | 46.03% | 34.70% | 39.80% | 35.52% |
| GoogLeNet | TAN | 81.04% | 99.09% | 66.59% | 56.72% | 63.86% | 55.02% |
| | MIFGSM | 61.56% | 98.36% | 47.57% | 34.16% | 37.57% | 29.75% |
| | DIFGSM | 58.81% | 94.47% | 47.91% | 32.17% | 36.20% | 26.88% |
| | NIFGSM | 31.46% | 64.32% | 25.34% | 19.85% | 23.14% | 19.63% |
| | SINIFGSM | 41.97% | 69.79% | 34.39% | 28.21% | 29.77% | 25.48% |
| | VMIFGSM | 53.37% | 97.84% | 42.19% | 30.67% | 34.94% | 26.36% |
| | VNIFGSM | 56.26% | 95.74% | 43.96% | 32.31% | 36.11% | 29.49% |
| InceptionV3 | TAN | 75.11% | 71.56% | 98.81% | 67.23% | 63.62% | 54.57% |
| | MIFGSM | 42.64% | 35.92% | 96.00% | 32.49% | 35.00% | 29.51% |
| | DIFGSM | 42.99% | 33.70% | 86.72% | 31.16% | 34.13% | 28.20% |
| | NIFGSM | 27.12% | 24.67% | 51.66% | 19.49% | 23.76% | 22.45% |
| | SINIFGSM | 26.76% | 25.23% | 62.46% | 21.90% | 24.36% | 22.59% |
| | VMIFGSM | 36.38% | 34.05% | 91.54% | 30.15% | 31.43% | 28.52% |
| | VNIFGSM | 37.82% | 33.55% | 84.02% | 31.44% | 32.28% | 28.58% |
| Mobilenet | TAN | 61.30% | 57.66% | 61.53% | 97.40% | 60.97% | 63.11% |
| | MIFGSM | 19.98% | 18.66% | 22.87% | 99.86% | 23.55% | 20.31% |
| | DIFGSM | 23.96% | 21.92% | 23.79% | 91.64% | 24.51% | 22.65% |
| | NIFGSM | 15.76% | 15.58% | 16.85% | 80.05% | 18.06% | 15.91% |
| | SINIFGSM | 16.81% | 15.52% | 18.96% | 85.14% | 21.20% | 16.63% |
| | VMIFGSM | 18.46% | 17.84% | 18.70% | 99.40% | 21.49% | 19.61% |
| | VNIFGSM | 21.60% | 18.41% | 22.34% | 95.58% | 24.67% | 21.96% |
| ResNet50 | TAN | 71.39% | 71.54% | 71.02% | 73.68% | 97.69% | 66.26% |
| | MIFGSM | 43.23% | 30.51% | 41.57% | 42.41% | 97.08% | 36.29% |
| | DIFGSM | 45.18% | 34.25% | 42.37% | 39.40% | 90.35% | 34.36% |
| | NIFGSM | 22.07% | 20.45% | 20.33% | 19.36% | 45.34% | 19.75% |
| | SINIFGSM | 25.81% | 21.38% | 27.15% | 31.01% | 71.64% | 26.02% |
| | VMIFGSM | 36.44% | 26.33% | 35.75% | 38.61% | 96.17% | 32.79% |
| | VNIFGSM | 40.80% | 27.10% | 38.26% | 38.87% | 94.17% | 36.49% |
| Shufflenet | TAN | 53.91% | 47.78% | 51.69% | 60.35% | 58.78% | 98.36% |
| | MIFGSM | 18.29% | 16.43% | 17.06% | 19.46% | 17.20% | 100.00% |
| | DIFGSM | 23.55% | 20.36% | 20.80% | 22.55% | 21.35% | 97.54% |
| | NIFGSM | 13.96% | 13.06% | 13.14% | 14.47% | 13.66% | 91.77% |
| | SINIFGSM | 15.83% | 15.23% | 15.34% | 19.42% | 16.05% | 95.79% |
| | VMIFGSM | 17.58% | 16.34% | 17.09% | 21.65% | 18.46% | 99.94% |
| | VNIFGSM | 19.43% | 17.97% | 18.68% | 22.87% | 19.98% | 99.73% |

In conclusion, for both non-targeted and targeted attacks, our approach generates adversarial examples with the strongest transferability. In other words, it performs better on exploring the common vulnerability of DNN models. We attribute this to the adversarial training between the generator and attenuator. Figuratively speaking, it is because of the attenuator constantly creating obstacles for the generator that the attack capability of the generator is continuously enhanced and completed.

### 4.7. Comparison of Real-Time Performance

According to (4), compared to traditional iterative methods, the generator in our approach is capable of one-step mapping original samples to adversarial examples. It acts like a function that takes inputs and outputs results based on the mapping relationship. To evaluate the real-time performance of adversarial attacks, we compared the time cost of generating a single adversarial example through different attack algorithms. The time consumption of non-targeted and targeted attacks is shown in Tables 8 and 9, respectively.

As we can see, there is almost no difference in the time cost of crafting a single adversarial example in non-targeted and targeted attacks. Meanwhile, for all the victim models, the time cost of generating a single adversarial example through our method is stable around 2 ms. As for baseline methods, it depends on the complexity of victim models, the more complex the model, the longer the time cost. However, even for the simplest victim model, the minimum time cost of baseline methods is about 4.5 ms, consuming twice as much time as our approach. Thus, there is no doubt that the proposed method achieves the most superior and stable real-time performance.

**Table 8.** Time cost of generating a single adversarial example through different attack algorithms in non-targeted attacks.

| Method | DenseNet121 | GoogLeNet | InceptionV3 | Mobilenet | ResNet50 | Shufflenet | Mean |
|--------|-------------|-----------|-------------|-----------|----------|------------|------|
| TAN | 0.002029 s | 0.002201 s | 0.002039 s | 0.002218 s | 0.002031 s | 0.002045 s | 0.002094 s |
| MIFGSM | 0.018285 s | 0.006351 s | 0.012636 s | 0.005093 s | 0.013445 s | 0.004451 s | 0.010044 s |
| DIFGSM | 0.018276 s | 0.006363 s | 0.012653 s | 0.005103 s | 0.013468 s | 0.004488 s | 0.010059 s |
| NIFGSM | 0.018312 s | 0.006354 s | 0.012646 s | 0.005111 s | 0.013477 s | 0.004456 s | 0.010059 s |
| SINIFGSM | 0.091032 s | 0.031499 s | 0.063015 s | 0.024865 s | 0.067202 s | 0.021676 s | 0.049882 s |
| VMIFGSM | 0.109252 s | 0.037827 s | 0.075580 s | 0.029803 s | 0.080479 s | 0.025968 s | 0.059818 s |
| VNIFGSM | 0.109184 s | 0.037804 s | 0.075483 s | 0.029776 s | 0.080560 s | 0.025907 s | 0.059786 s |

**Table 9.** Time cost of generating a single adversarial example through different attack algorithms in targeted attacks.

| Method | DenseNet121 | GoogLeNet | InceptionV3 | Mobilenet | ResNet50 | Shufflenet | Mean |
|--------|-------------|-----------|-------------|-----------|----------|------------|------|
| TAN | 0.002070 s | 0.002069 s | 0.002036 s | 0.002055 s | 0.002087 s | 0.002097 s | 0.002069 s |
| MIFGSM | 0.018281 s | 0.006353 s | 0.012634 s | 0.005088 s | 0.013451 s | 0.004446 s | 0.010042 s |
| DIFGSM | 0.018291 s | 0.006369 s | 0.012652 s | 0.005104 s | 0.013490 s | 0.004488 s | 0.010065 s |
| NIFGSM | 0.018306 s | 0.006358 s | 0.012661 s | 0.005105 s | 0.013486 s | 0.004460 s | 0.010063 s |
| SINIFGSM | 0.091064 s | 0.031539 s | 0.063066 s | 0.024871 s | 0.067216 s | 0.021664 s | 0.049903 s |
| VMIFGSM | 0.109262 s | 0.037860 s | 0.075579 s | 0.029776 s | 0.080481 s | 0.025984 s | 0.059823 s |
| VNIFGSM | 0.109176 s | 0.037819 s | 0.075502 s | 0.029798 s | 0.080546 s | 0.025923 s | 0.059794 s |

### 4.8. Visualization of Adversarial Examples

In this section, we took ResNet50 as the surrogate model and visualized the adversarial examples crafted by different methods in Figure 9. Obviously, the adversarial perturbations generated by our method are continuous, and mainly focus on the target region of SAR images. In contrast, the perturbations produced by baseline methods are quite discrete, and almost cover the global area of SAR images. First, from the perspective of feature extraction, since the features that have a greater impact on recognition results are mainly concentrated in the target region rather than the background clutter area, a focused disruption of key features is certainly a more efficient attack strategy. Second, from the perspective of physical feasibility, the fewer pixels modified in adversarial examples, the smaller range perturbed in reality, so localized perturbations are more feasible than global ones. In summary, the proposed method improves the efficiency and feasibility of adversarial attacks by focusing perturbations on the target region of SAR images.

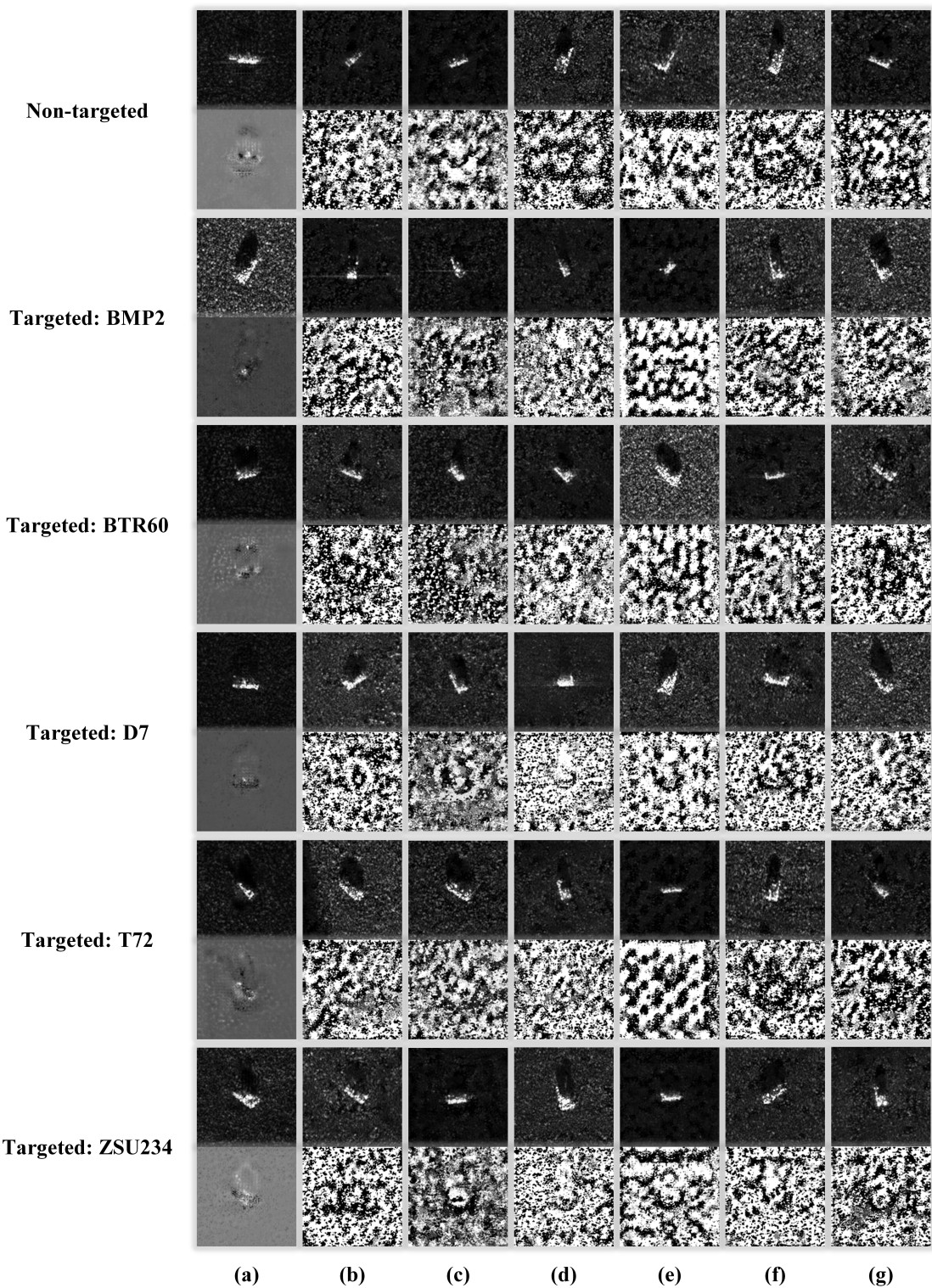

**Figure 9.** Visualization of adversarial examples against ResNet50. (**a**) TAN. (**b**) MIFGSM. (**c**) DIFGSM. (**d**) NIFGSM. (**e**) SINIFGSM. (**f**) VMIFGSM. (**g**) VNIFGSM. From top to bottom, the corresponding target classes are None, BMP2, BTR60, D7, T72, and ZSU234, respectively. For each attack, the first row shows adversarial examples, and the second row shows corresponding adversarial perturbations.

## 5. Discussions

So far, the proposed method has been proven to be effective for SAR images. Further studies should verify its effectiveness in other fields, such as optical [40,41], infrared [42,43],

and synthetic aperture sonar (SAS) [44–47] images, etc. Although different imaging principles lead to huge differences in the resolution, dimension, and target type of images, we argue that TAN can be well-suitable to these fields. The reason is that adversarial examples essentially attack the inherent vulnerability of DNN models, independent of the input data. However, the non-negligible challenge is how to realize these adversarial examples in the real world. Specifically, the physical implementation depends on the imaging principle, e.g., crafting adversarial patches against optical cameras, changing temperature against infrared devices, and emitting acoustic signals against SAS, etc. This is a worthwhile topic in the future.

### 6. Conclusions

This paper proposed a transferable adversarial network (TAN) to attack DNN-based SAR-ATR models, with the benefit that the transferability and the real-time performance of adversarial examples is significantly improved, which is of great significance for real-world black-box attacks. In the proposed method, we simultaneously trained two encoder–decoder models: a generator that learns the one-step forward mapping from original data to adversarial examples, and an attenuator that captures the most harmful deformations to malicious samples. It is motivated by enabling real-time attacks by one-step mapping original data to adversarial examples, and enhancing the transferability through a two-player game between the generator and attenuator. Experimental results demonstrated that our approach achieves state-of-the-art transferability with acceptable adversarial perturbations and minimum time costs compared to existing attack methods, making real-time black-box attacks without any prior knowledge a reality. Potential future work could consider attacking DNN-based SAR-ATR models under small sample conditions. In addition to improving the performance of attack algorithms, it makes sense to implement adversarial examples in the real world.

**Author Contributions:** Conceptualization, M.D. (Meng Du) and D.B.; methodology, M.D. (Meng Du); software, M.D. (Meng Du); validation, D.B., Y.S., B.S. and Z.W.; formal analysis, D.B. and M.D. (Mingyang Du); investigation, M.D. (Mingyang Du); resources, D.B.; data curation, M.D. (Meng Du); writing—original draft preparation, M.D. (Meng Du); writing—review and editing, M.D. (Meng Du), L.L. and D.B.; visualization, M.D. (Meng Du); supervision, D.B.; project administration, D.B.; funding acquisition, D.B. All authors have read and agreed to the published version of the manuscript.

**Funding:** This work was supported by the National Natural Science Foundation of China under Grant 62071476.

**Institutional Review Board Statement:** The study does not involve humans or animals.

**Informed Consent Statement:** The study does not involve humans.

**Data Availability Statement:** The experiments in this paper use public datasets, so no data are reported in this work.

**Conflicts of Interest:** The authors declare that they have no conflict of interest to report regarding the present study.

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
