# Peer review of "TAN: A Transferable Adversarial Network for DNN-Based UAV SAR Automatic Target Recognition Models"

_drones, doi:10.3390/drones7030205_

Round 1

Reviewer 1 Report

In this paper, the authors discuss a transferable adversarial network (TAN) which utilizes the encoder-decoder model and data augmentation technology to improve the transferability and real-time performance of adversarial examples. In general, the work in this paper wothy publication. However, there are some queries the authors should be addressed.

1. TAN introduces a generator G(·) to learn the one-step forward mapping from the clean sample x to the adversarial example. The results of this step should be presented and discussed in experiments to verify its effectiveness

2.  In Eq. (9), Eq. (13) and Eq. (17), the linear weighted sum method is used. The reviewer wanders to know how to determine the weight coefficients. From reviewer’s view, the cost function is needed to compute weight coefficients.

3. The synthetic aperture technique is further used in underwater field, and it is named synthetic aperture sonar (SAS), which can also be mounted in Unmanned Undersea Vehicles. Since the SAS image [1-4] is similar to SAR image, the reviewer wanders to know whether the authors’ method can be suitable for SAS image. Besides, the reviewer wanderers to know the usage challenge and major difference of authors’ method in SAS image and SAR image. This discussion would be very helpful for the technique field.

[1] Chou-Wei Kiang, et al.Imaging on Underwater Moving Targets With Multistatic Synthetic Aperture Sonar.IEEE Transactions on Geoscience and Remote Sensing,2022,60:4211218.

[2]X. Zhang,et al.Multireceiver SAS imagery based on monostatic conversion.IEEE Journal of Selected Topics in Applied Earth Observations and Remote Sensing,2021,14:10835-10853.

[3] Ha-min Choi, et al.Compressive Underwater Sonar Imaging with Synthetic Aperture Processing.Remote Sensing,2021,13:1924.

[4] David J. Pate, et al.Estimation of Synthetic Aperture Resolution by Measuring Point Scatterer Responses.IEEE Journal of Oceanic Engineering,2022,47:457 - 471.

4. The weight coefficients in the first paragraph of section 4.2 should be explained the reason, as this paper does not discuss how to determine the weight coefficients.

Author Response

Response to Reviewer 1 Comments

Point 1: TAN introduces a generator G(·) to learn the one-step forward mapping from the clean sample x to the adversarial example. The results of this step should be presented and discussed in experiments to verify its effectiveness.

Response 1: We thank the reviewer for pointing this out. The discussion of this step has been added to Section 4.7:

According to formula 4, compared to traditional iterative methods, the generator in our approach is capable of one-step mapping original samples to adversarial examples. It acts like a function that takes inputs and outputs results based on the mapping relationship. To evaluate the real-time performance of adversarial attacks, we compared the time cost of generating a single adversarial example through different attack algorithms. The time consumption of non-targeted and targeted attacks is shown in Tables 8 and 9, respectively. (See lines 399-404)

Point 2: In Eq. (9), Eq. (13) and Eq. (17), the linear weighted sum method is used. The reviewer wanders to know how to determine the weight coefficients. From reviewer’s view, the cost function is needed to compute weight coefficients.

Response 2: Thank you for your valuable suggestion. We provide the following explanations for your confusion:

The weight coefficients represent the relative importance of each loss term during the training process. A larger weight implies that the corresponding loss will decrease more rapidly and significantly, allowing attackers to adjust the parameters flexibly according to their actual needs. (See lines 181-184)

Point 3: The synthetic aperture technique is further used in underwater field, and it is named synthetic aperture sonar (SAS), which can also be mounted in Unmanned Undersea Vehicles. Since the SAS image is similar to SAR image, the reviewer wanders to know whether the authors’ method can be suitable for SAS image. Besides, the reviewer wanderers to know the usage challenge and major difference of authors’ method in SAS image and SAR image. This discussion would be very helpful for the technique field.

Response 3: Thank you so much for your valuable comment, and we have read a large amount of relevant literature to expand and discuss our research, as detailed in Section 5:

So far, the proposed method has been proven to be effective for SAR images. Further studies should verify its effectiveness in other fields, such as optical [40,41], infrared [42,43], and synthetic aperture sonar (SAS) [44-47] images, etc. Although different imaging principles lead to huge differences in the resolution, dimension, and target type of images, we argue that TAN can be well-suitable to these fields. The reason is that adversarial examples essentially attack the inherent vulnerability of DNN models, independent of the input data. However, the non-negligible challenge is how to realize these adversarial examples in the real world. Specifically, the physical implementation depends on the imaging principle, e.g., crafting adversarial patches against optical cameras, changing temperature against infrared devices, and emitting acoustic signals against SAS, etc. This is a worthwhile topic in the future. (See lines 427-437)

Point 4: The weight coefficients in the first paragraph of section 4.2 should be explained the reason, as this paper does not discuss how to determine the weight coefficients.

Response 4: We gratefully appreciate your rigorous advice, and the way we determine the parameters is as follows:

For better attack performance, the hyperparameters of TAN are fine-tuned through numerous experiments, and the following set of parameters is eventually determined to best meet our requirements. Specifically, we set the generator loss weights [wG1, wG2, wG3] to [0.25, 0.25, 0.5], the attenuator loss weights [wA1, wA2, wA3] to [0.25, 0.25, 0.5], the training ratio to 3, the training epoch to 50, and the batch size to 8. (See lines 264-269)

Reviewer 2 Report

A transferable adversarial network for DNN-based UAV SAR automatic target recognition models is proposed. It is interesting but not complete enough for publication. Specific comments are listed as follows:

(1) Some spelling mistakes and writing style can be further improved, for example: Line 172 at page 6: “between LG1, LG2, and LG3.”, these small mistakes need to be corrected before the paper is published.

(2) The current abstract cannot easily make people understand the novelty and significance of the paper. Therefore, the abstract needs to be corrected so that the reader can understand innovation and value of this paper more easily.

(3) Some references are too old in introduction section. The authors need to refer to some of the latest methods and literature to replace them.

(4) Transferability of Adversarial Examples at page 3: why the similarity between models, adversarial examples generated for a certain model can also successfully attack other models performing the same task? As an important theoretical basis, the authors should elaborate on it in the paper.

(5) Formula (8) at page 5: Whether the parameter p is used as a training parameter or a specified value?

(6) Line 256 at page 9: what does “and the norm type to the L2-norm.” mean? If mean p equals 2, what is the basis choosing p equals 2? The authors should elaborate on it in the paper.

(7) Line 257 at page 10: Why do you use Adam optimization method? Why not use other methods, such as stochastic gradient descent (SGD)? What are the advantages of Adam?

(8) Table 8 and table 9 at page 17: time cost unit needs to be indicated.

(9) It is suggested to correct the conclusion to highlight the most important results and the significance of the research.

(10) In the introduction section, the author omitted some important references about SAR ATR methods based on deep learning, such as

1)   https://www.doi.org/10.1109/JSTARS.2022.3141485

2)   https://www.doi.org/10.1109/TGRS.2023.3248040

3)   https://www.doi.org/10.1109/LGRS.2018.2877599

Author Response

Response to Reviewer 2 Comments

Point 1: Some spelling mistakes and writing style can be further improved, for example: Line 172 at page 6: “between LG1, LG2, and LG3.”, these small mistakes need to be corrected before the paper is published.

Response 1: We feel sorry for the language errors in this paper. The manuscript has been carefully revised by a native English speaker to fix the grammatical and sentence errors.

Point 2: The current abstract cannot easily make people understand the novelty and significance of the paper. Therefore, the abstract needs to be corrected so that the reader can understand innovation and value of this paper more easily.

Response 2: We completely agree with you. To help readers better understand the innovation and value of this paper, the abstract has been modified as follows:

Recently, the unmanned aerial vehicle (UAV) synthetic aperture radar (SAR) has become a highly sought-after topic for its wide applications in the target recognition, detection, and tracking. However, SAR automatic target recognition (ATR) models based on deep neural networks (DNN) are suffering from adversarial examples. Generally, non-cooperators rarely disclose any SAR-ATR model information, making adversarial attacks challenging. To tackle this issue, we propose a novel attack method called Transferable Adversarial Network (TAN). It can craft highly transferable adversarial examples in real time and attack SAR-ATR models without any prior knowledge, which is of great significance for real-world black-box attacks. The proposed method improves the transferability via a two-player game, in which we simultaneously train two encoder-decoder models: a generator that crafts malicious samples through a one-step forward mapping from original data, and an attenuator that weakens the effectiveness of malicious samples by capturing the most harmful deformations. Particularly, compared to traditional iterative methods, the encoder-decoder model can one-step map original samples to adversarial examples, thus enabling real-time attacks. Experimental results indicate that our approach achieves state-of-the-art transferability with acceptable adversarial perturbations and minimum time costs compared to existing attack methods, making real-time black-box attacks without any prior knowledge a reality. (See lines 1-16)

Point 3: Some references are too old in introduction section. The authors need to refer to some of the latest methods and literature to replace them.

Response 3: Thank you for pointing this out. We have replaced some of the non-critical references with the latest studies in five years, such as references 4, 5, and 6.

Point 4: Transferability of Adversarial Examples at page 3: why the similarity between models, adversarial examples generated for a certain model can also successfully attack other models performing the same task? As an important theoretical basis, the authors should elaborate on it in the paper.

Response 4: We feel sorry for the confusion brought to the reviewer. In fact, after careful deliberation, we believe the original statement is inaccurate. Thus, we review reliable literature [18] and find that the transferability of adversarial examples is demonstrated by extensive experiments. The relevant statement has been modified as follows:

Specifically, the extensive experiments in [18] have demonstrated that adversarial examples can transfer among models, even if they have different architectures or are trained on different training sets, so long as they are trained to perform the same task. Details about the transferability are shown in Figure 1. (See lines 133-136)

Point 5: Formula (8) at page 5: Whether the parameter p is used as a training parameter or a specified value?

Response 5: Thank you for your rigorous comment. The parameter p is not a training parameter, but a hyperparameter of our method. The detailed description is as follows:

The common Lp-norm includes the L0-norm, L2-norm, and Linf-norm. Attackers can select different norm types according to practical requirements. For example, the L0-norm represents the number of modified pixels in adversarial examples, the L2-norm measures the mean square error (MSE) between adversarial examples and original samples, and the Linf-norm denotes the maximum variation for individual pixels in adversarial examples. (See lines 109-113)

Point 6: Line 256 at page 9: what does “and the norm type to the L2-norm.” mean? If mean p equals 2, what is the basis choosing p equals 2? The authors should elaborate on it in the paper.

Response 6: Thank you for pointing this out. We combine response 5 and clarify the reason as follows:

During the training phase of TAN, to minimize the MSE between adversarial examples and original samples, we adopt the L2-norm to evaluate the image distortion caused by adversarial perturbations. (See lines 109-113 and 262-264)

Point 7: Line 257 at page 10: Why do you use Adam optimization method? Why not use other methods, such as stochastic gradient descent (SGD)? What are the advantages of Adam?

Response 7: We appreciate you pointing this out, and the reasons for selecting the Adam optimizer are as follows:

Due to the adversarial process involved in TAN, training can be challenging to converge. As such, we employ Adam [38], a more computationally efficient optimizer, to accelerate model convergence, which also performs better in solving non-stationary objective and sparse gradient problems. The learning rate is set to 0.001. (See lines 269-272)

Point 8: Table 8 and table 9 at page 17: time cost unit needs to be indicated.

Response 8: Thank you for your rigorous comment, and we have added time units (s) to all the data in Tables 8 and 9.

Point 9: It is suggested to correct the conclusion to highlight the most important results and the significance of the research.

Response 9: We completely agree with you, and the conclusion has been revised as follows:

This paper proposed a transferable adversarial network (TAN) to attack DNN-based SAR-ATR models, with the benefit that not only the transferability but also the real-time performance of adversarial examples is significantly improved, which is of great significance for real-world black-box attacks. In the proposed method, we simultaneously trained two encoder-decoder models: a generator that learns the one-step forward mapping from original data to adversarial examples, and an attenuator that captures the most harmful deformations to malicious samples. It is motivated by enabling real-time attacks by one-step mapping original data to adversarial examples, and enhancing the transferability through a two-player game between the generator and attenuator. Experimental results demonstrated that our approach achieves state-of-the-art transferability with acceptable adversarial perturbations and minimum time costs compared to existing attack methods, making real-time black-box attacks without any prior knowledge a reality. Potential future work could consider attacking DNN-based SAR-ATR models under small sample conditions. In addition to improving the performance of attack algorithms, it makes sense to implement adversarial examples in the real world. (See lines 439-453)

Point 10: In the introduction section, the author omitted some important references about SAR ATR methods based on deep learning.

Response 10: Thank you for pointing this out. We have added these important efforts to our paper, see references 4, 5, and 6 for details.

Round 2

Reviewer 1 Report

This paper has been improved after revision, and can give readers much insightful ideas. 

Reviewer 2 Report

The authors have revised the manuscript according to the comments of reviewers. I have no other questions.